# Photoemission-based microelectronic devices

Ebrahim Forati[1], Tyler J. Dill[2], Andrea R. Tao[2] & Dan Sievenpiper[1]

The vast majority of modern microelectronic devices rely on carriers within semiconductors due to their integrability. Therefore, the performance of these devices is limited due to natural semiconductor properties such as band gap and electron velocity. Replacing the semiconductor channel in conventional microelectronic devices with a gas or vacuum channel may scale their speed, wavelength and power beyond what is available today. However, liberating electrons into gas/vacuum in a practical microelectronic device is quite challenging. It often requires heating, applying high voltages, or using lasers with short wavelengths or high powers. Here, we show that the interaction between an engineered resonant surface and a low-power infrared laser can cause enough photoemission via electron tunnelling to implement feasible microelectronic devices such as transistors, switches and modulators. The proposed photoemission-based devices benefit from the advantages of gas-plasma/vacuum electronic devices while preserving the integrability of semiconductor-based devices.

[1] Electrical and Computer Engineering Department, University of California San Diego, La Jolla, California 92098-0407, USA. [2] Department of NanoEngineering, University of California San Diego, La Jolla, California 92098-0448, USA. Correspondence and requests for materials should be addressed to E.F. (email: forati@ieee.org) or to D.S. (email: dsievenpiper@eng.ucsd.edu).

In 1906, the first vacuum-based electronic device, a diode, was invented by Fleming, and later in 1907 Lee De Forest introduced the first vacuum-based amplifier. Low-pressure gas was then added to the vacuum tubes to increase their power handling due to the excess current generation by the ionized gas. During the 1960s and 1970s, vacuum and gas-plasma electronic devices such as voltage regulators, switches and modulators were widely used in radio frequency (RF) communication and audio systems. After being mostly replaced by semiconductor counterparts, due to their integrability, research on vacuum electronic devices was mostly directed towards high-power travelling wave tubes and Terahertz (THz) sources. In addition, gas-plasma devices (usually with micro-scale dimensions) have been studied for plasma displays, water treatment, ozone generation, pollution control, medical treatment and material processing[1–4].

On the other hand, further optimization of semiconductor devices is becoming more challenging due to the limitations of the natural properties of semiconductors such as bandgap and electron mobility. For some applications, replacement of semiconductors with substitute materials may open up new opportunities for scaling characteristics of existing electronic devices such as the speed, power, and so on. For instance, vacuum or gas plasma devices benefit from higher mobility of electrons than their semiconductor counterparts. As an example, the electron mobility under an electric field strength of $10^3 \, \mathrm{V \, cm^{-1}}$ in neon gas (at pressure 100 torr and temperature 300 K leading to the atomic density of $[\mathrm{Ne}] = 3.2 \times 10^{18} \, \mathrm{cm^{-3}}$) is greater than $10^4 \, \mathrm{cm^2 \, V^{-1} \, s^{-1}}$. This mobility is $\sim 7$ times larger than the electron mobility in silicon (Si) at 300 K (refs 5,6). However, issues such as gas plasma ignition (which requires high static voltages or high laser intensities), electrode erosion due to gas atom collisions in plasma, electron injection into vacuum (typically by thermoionic emission), and the lack of integrability with other (semiconductor) micro-devices have reduced development of micro-plasma and micro-vacuum devices to compete with semiconductor microelectronics.

The main mechanisms which extract electrons from a material (mostly metals) are: thermoionic emission, electric field emission, the photoelectric effect and photoemission. In a thermoionic emission process, electrons are transferred over the surface potential barrier of the metal (the work function) due to the added thermal energy. Most of the vacuum electronic devices, including vintage triodes and modern magnetrons, rely on thermoionic emission. However, thermionic emission requires cathode temperatures on the order of 1,000 K which makes it infeasible in micro-scale dimensions. Electric field emission, also called Fowler–Nordheim tunnelling, is the process whereby electrons tunnel through the reduced work function due to the presence of a high electric field (typically static). This has been investigated for realizing cold cathode emitters to replace thermoionic emitters, since they do not require high power to thermally extract electrons from the surface, yet they require high bias voltages (for example, close to 100 V in ref. 7)[8,9]. A general definition of the photoelectric effect is electron transition to a higher energy level due to excitation by a photon[10]. Depending on the photon energy and number, electrons may be excited internally, leading to photoconductivity, or leave the metal, leading to electron emission. If the photon energy is larger than the material's work function, photons can couple to electrons effectively and transfer them over the work function. This process, which was discovered by Heinrich Hertz in 1887 (ref. 11), is independent of the number of photons, and is used in designing vacuum/gas-filled phototubes and photo-multiplier tubes, some of which were eventually superseded by semiconductor photo-resistors and photo-diodes

(photo-multiplier tubes are still in use). However, photons with an energy lower than the material's work function can also liberate electrons by either tunnelling electrons through the potential barrier, or transferring them over the barrier (multi-photon absorption). This process requires a high number of photons, and is called photoemission in some works, as we do in the rest of this work[10,12–16]. Unlike Hertz's experiments, the photon energy in photoemission is less than the unperturbed metal work function, and the key factor is the laser–matter interaction (either strong-field or perturbative), caused by the nanolocalized electromagnetic field in the vicinity of metallic structures (such as sharp metallic nanotapers). Typically, laser intensities on the order of $1 \, \mathrm{TW \, cm^{-2}}$ (in the infrared (IR) range) are required for photoemission[17]. However, photoemission can be greatly enhanced by the excitation of collective electron modes of the metal, called surface plasmon polaritons (SPPs), and laser intensities on the order of $1 \, \mathrm{GW \, cm^{-2}}$ in the mid-IR have been shown to be enough for photoemission[18]. In photomultiplier tubes[19] and field emitter arrays[7,20,21], a static bias is typically used to decrease the potential barrier of the material, enabling it to emit electrons with light. This combination can provide great sensitivity (single photon in case of a photomultiplier tube) or high electron emission (bright electron beam in case of field emitter arrays). Historically, these emission processes have been mostly studied for vacuum, and well-established theories exist to estimate their current densities. However, the same conclusions can be used at higher pressures for dimensions smaller than the mean free path of electrons (about a few hundreds of nm in air[22–25]).

Here, we propose to use the combination of photoemission (assisted by localized surface plasmon resonances (LSPRs)) and field emission to inject electrons into the surrounding space (vacuum or gas) and therefore to realize semiconductor-free microelectronic devices such as switches, transistors, photo-detectors, and so on. Other applications include photocathode source for accelerators (for example, linear accelerators) and free electron lasers, as well as a source for higher harmonic generation. We show that, by exciting LSPRs, unprecedented laser intensities of around $1 \, \mathrm{W \, cm^{-2}}$ along with a small bias voltage ($< 10 \, \mathrm{V}$) can activate a semiconductor-free device. Due to their small dimensions (sub-micrometer scale) and fabrication method (lift-off process), the proposed photoemission-based devices can also be integrated with semiconductor devices.

## Results

**The resonant surface design and the device implementation.** We fabricated an engineered micro-surface which supports LSPRs in the near-IR range, and enabled us to apply static voltage between inclusions of the surface. In our scheme, combined photonic and electric excitation of a metallic micro-surface causes electron emission and acceleration into the surrounding space. External electric or magnetic fields can then be applied to guide or manipulate these electrons for different device realizations. Figure 1 depicts our designed two-port device to study the photoemission simultaneously with an applied static bias. In the designed device, electron emission occurs at the high electric field spots between the resonant inclusions (due to both LSPR and the static bias). The intensity of the electric field at the hot spots can be controlled both electrically (with static bias) and optically (with the incoming laser). We will show that the two isolated ports in Fig. 1 can couple together due to the free electrons caused by photoemission.

The design was inspired by surface-enhanced Raman scattering (SERS) in which the local electric field is greatly enhanced due to

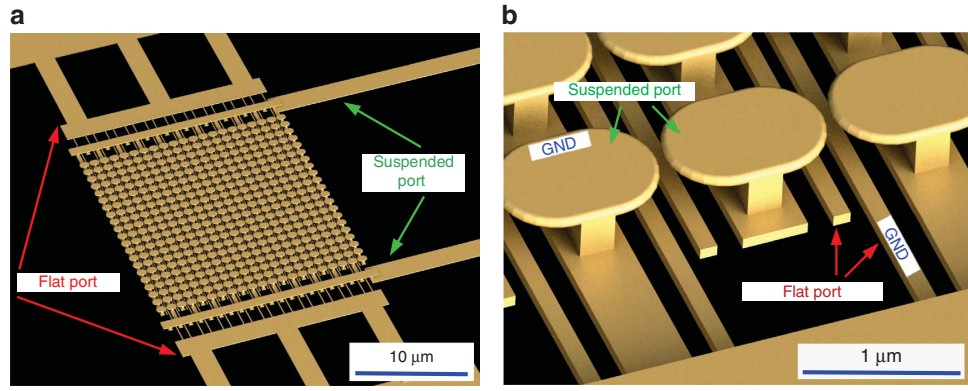

**Figure 1 | The designed photoemission-based device.** (**a**) Biased resonant inclusions of each port under illumination by a CW laser can emit electrons, (**b**) the free electrons can be manipulated electrically by applying voltage $V_f$ on the flat port and $V_s$ on the suspended port. The grounded (GND) terminal of each port is specified.

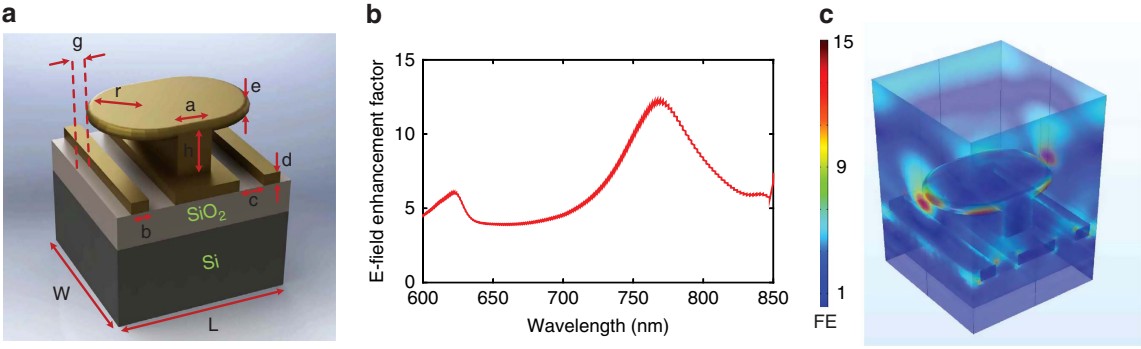

**Figure 2 | The designed resonant surface.** (**a**) Dimensions of the unit cell are a = 100 nm, b = 100 nm, c = 150 nm, d = 80 nm, e = 70 nm, g = 50 nm, r = 240 nm, L = 850 nm, W = 880 nm, h = 225 nm, (**b**) full wave (ANSYS HFSS) simulated electric field enhancement at the center of the gap between the inclusions, and (**c**) the electric field magnitude distribution at $\lambda = 785$ nm (red colour represents the highest value). This figure only shows the laser-surface interaction, and there are not any bias voltages involved.

the surface roughness to amplify the Raman response of biomolecules (Supplementary Figs 1 and 2)[6,26,27]. The intention in the design of the device was to exploit the distributed high quality factor (Q) resonance, inherent in certain periodic structures, to dramatically enhance the absorption of local photons and facilitate photoemission[28–30].

Figure 2 shows the unit cell of the high Q resonant surface which we used to electro-optically emit electrons. The unit cell consists of gold metallic inclusions on a silicon wafer with a layer of silicon dioxide ($SiO_2$) in between as isolation. The metallic inclusions consists of a vertical gold post topped with a gold plate, called the mushroom, along with ribbons on the substrate. Silicon wafers with a layer of $SiO_2$ (typically between 100 and 600 nm) are usually used as the substrate in photo-detection devices. The $SiO_2$ layer is used as an isolator to minimize the leakage current in the device. Usually a 200 nm thick layer of $SiO_2$ provides enough isolation[31]. The Si wafers used in our experiments had 1,000 Ω cm resistivity and the $SiO_2$ layer was coated on the wafer using plasma sputtering. Full wave simulation of the unit cell, Fig. 2b, confirms a resonance at $\lambda = 785$ nm with the electric field enhancement (FE) of about FE = 12 (defined as the ratio of the maximum to the incident electric field at the gap center) under proper linear polarization (along the mushroom's length). The field enhancement is due to the localized surface plasmon resonance supported by gold[27,32–36]. The resonant mode was optimized so that the enhanced electric field at resonance

(hot spot) is confined to the gap between mushrooms as Fig. 2c shows. As a result, the maximum static electric field (due to the bias), is superimposed with the laser-induced hot spot. Nonetheless, the flat port (as defined in Fig. 1b) also experiences field enhancement of about half of the maximum FE as shown in Fig. 2c, which will be shown later that is sufficient to emit electrons.

The unit cell in Fig. 2 also provides two electrical ports. The suspended electrical port consists of mushrooms, while the second electrical port is formed by the gold ribbons on the substrate, as shown in Fig. 1b, and is called the flat port. With this generic design, we will quantify an important coupling parameter between the two ports, that is, transconductance, due to photoemission. Figure 3 shows scattering electron microscopy (SEM) pictures of the fabricated device including an array of 21 by 21 unit cells connected to four large square pads (250 μm²) for wire bonding. To form the suspended port, so that the static electric field hot spots lay inside the gap between the mushrooms, we needed to feed every row of mushrooms with alternate polarities. This was done by placing two air bridges on the surface sides, and connecting them to the wire bonding square pads, as clarified in Fig. 3 (Supplementary Fig. 3). As a result, after biasing the suspended port, adjacent mushroom rows will have opposite polarities, similar to an inter-digital capacitor. The ribbons on the substrate were also connected to the remaining two square pads, forming the flat port. The surface was fabricated using a multi-

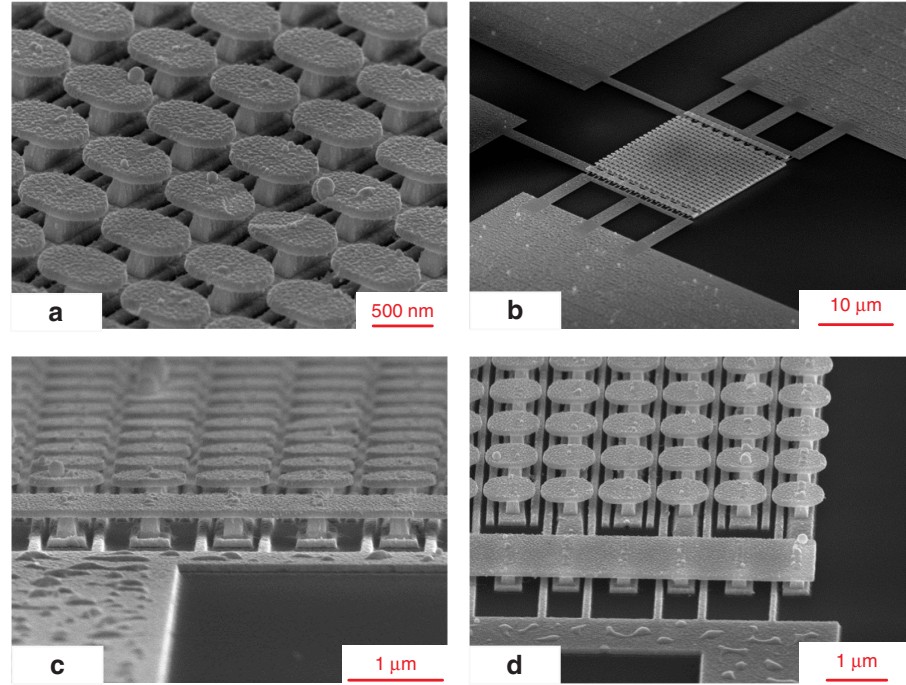

**Figure 3 | SEM pictures of the fabricated electron emission-based device.** (**a**) The resonant surface, (**b**) the entire device including the wire-bonding pads, (**c**) the airbridges on the two sides are for biasing mushroom rows with alternating polarities, to form the suspended port, (**d**) the parallel strips on the substrate, below the mushrooms, form the flat port.

| Table 1 \| The average and the standard deviation (s.d.) of the FE. | | | | |
|---|---|---|---|---|
| Sample No. | $\lambda = 633\,nm$ $FE_{ave}$ | $\lambda = 633\,nm$ s.d. | $\lambda = 785\,nm$ $FE_{ave}$ | $\lambda = 785\,nm$ s.d. |
| 1 | — | — | 25.80 | 0.90 |
| 2 | — | — | 23.00 | 0.97 |
| 3 | — | — | 24.78 | 0.98 |
| 4 | 13.10 | 1.38 | 27.51 | 1.38 |

step Ebeam lithography technique, as discussed in the methods section. To confirm the high $Q$ resonance of the surface, four different fabricated samples were characterized using Raman spectroscopy. For each sample, the electric field enhancement factor was measured at 15 different locations (using a stripe diode with dimensions 1.6 by 16.95 µm). Their averages and standard deviations are reported in Table 1. Details of the Raman spectroscopy and field enhancement determination is reported in the methods section. Based on the full-wave simulation results, the ratio of the maximum electric field (at the hot spot) to the average resonance-enhanced electric field on the surface is 1.24. Therefore, the maximum FE of surfaces can be approximated by multiplying the average FEs reported in Table 1 by a factor of 1.24. Based on the results in Table 1, the average maximum field enhancement of the samples was around $30 \times 1.24$, which provides substantial photoemission if combined with an applied static bias, as will be shown later. Obtaining very large enhancement factors is challenging due to metallic loss, as discussed in refs 37 and 38. To verify resonance, the FE of one sample was measured off resonance (at $\lambda = 633\,nm$) which was almost half of the resonant FE, as reported in Table 1. This is consistent with the full wave simulation result, shown in Fig. 2 as well. Reasons for observing higher experimental FEs

than simulation results include surface roughness, chemical enhancement and non-linearity of gold polarizability.

The fabricated samples were then installed and wire bonded inside standard dual in-line packages, as shown in the Supplementary Fig. 4.

**Individual and mutual port responses.** As the first experiment, the conductivity change of the suspended and flat ports are measured and reported in Fig. 4a. In all of the experiments of Fig. 4, one port is left open-circuited while measuring the other port. This was merely done to help us understand the physics of the device better. Also, except in Fig. 4c, throughout this paper the wavelength is set to be $\lambda = 785\,nm$. From Fig. 4a, it is evident that the optical port illumination changes the conductivity of the suspended and flat ports sufficiently to realize ON and OFF states, that is, the structure performs as an optical switch. The change in the conductivity is caused by the photoemitted electrons from the resonant inclusions on the surface, combined with static field emission at higher bias voltages. Based on the current versus voltage ($I$–$V$) curves on Fig. 4a, the conductivity of the suspended port increases by a factor of 10 after illumination by $I = 5\,\mathrm{W\,cm^{-2}}$ laser power at $\lambda = 785\,nm$ (with the 10 volts bias). Without applying a static voltage, the photoemission from the two inclusions of a port (either the flat or the suspended port) should be equal due to symmetry. Therefore, we expected a zero net current flow in the un-biased device's ports after laser illumination. However, there is some current flow in the flat port after laser illumination (even with zero voltage on the flat port) which we suspect is due to some asymmetry in the flat port fabrication. The magnitude of this photoemission current ($I_f$ while $V_f = 0$) increases from 100 to 800 nA as the laser intensity increases from $5\,\mathrm{W\,cm^{-2}}$ to $I = 40\,\mathrm{W\,cm^{-2}}$. The possible asymmetry due to fabrication means one terminal of the flat port can emit electrons more efficiently than the other terminal. This can explain the asymmetry in the $I$–$V$ curve of the flat port in Fig. 4a as well.

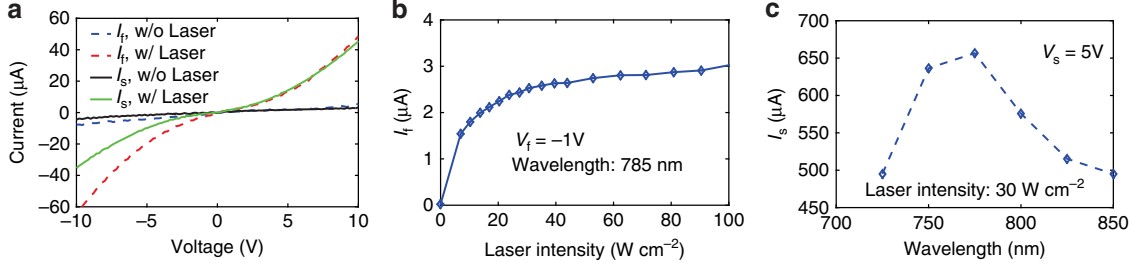

**Figure 4 | Individual port responses.** (**a**) I–V curves of the suspended/flat port as the flat/suspended port is open-circuited ($I = 5\,\mathrm{W\,cm^{-2}}$), (**b**) experimental responsivity of the flat port (markers) and their first order interpolation (line). $V_f$ and $I_f$ are voltage and current of the flat port, respectively. (**c**) Frequency dependence of the suspended port's response. Markers show the measured points, and the line is their first order interpolation. $V_s$ and $I_s$ are voltage and current of the suspended port, respectively.

The laser wavelength and intensity in this experiment were $\lambda = 785\,\mathrm{nm}$ and $I = 5\,\mathrm{W\,cm^{-2}}$, respectively, which are easily achievable with a low cost diode laser. Throughout the experiments, we set the pressure in the $10^{-4}$ torr range, to prevent any gas plasma formation around the device due to the static bias or laser illumination. That is, the device is placed inside a vacuum chamber with some electrical feedthroughs and optical view ports. This ensures that electron emission is the prevalent mechanism in the device as opposed to gas ionization. However, all of the results reported in this manuscript were also observed at atmospheric pressure (with slight differences). More specifically, at lower pressures, the conductivity due to the incoming photons is slightly higher than at air pressure, and the I–V curves are smoother. This is consistent with our expectations at lower pressures due to the reduced scattering of emitted electrons by gas atoms.

The emitted electrons can be manipulated by external electric or magnetic fields. In our design, this can be studied by measuring the mutual response between the suspended and flat ports, as summarized in Fig. 5. The I–V curve of the flat port with different applied voltages on the suspended port are shown in Fig. 5a. Since $I_f$ is created by the electron emission due to both photoemission and electric field emission, applying a bias voltage on the suspended port with similar or opposite polarity as the flat port increases or decreases the photoemitted current, respectively. For example, without any bias voltage on the suspended port (open circuit), and with $V_f = 10\,\mathrm{V}$, the laser illumination changes the flat port's conductivity by a factor of 10 (based on Fig. 4a). This conductivity change factor increases to 30 or decreases to 2, with applied bias voltages of $+10$ and $-10$ on the suspended port, respectively. Figure 5a demonstrates successful control of the flat port by both optical and suspended ports, which resembles a (semiconductor-free) transistor.

To quantify the mutual response of the two ports, the flat port was short circuited and its current was measured as a function of both the suspend port's voltage ($V_s$) and the laser power intensity, as shown in Fig. 5b. Another important information which Fig. 5b carries is the rate of the change in $I_f$ as $V_s$ varies (that is, the slope of the curves in Fig. 5b). This parameter which can be considered as the small-signal transconductance of the device, is shown in Fig. 5c. Although the device is not optimally designed for this purpose, Fig. 5c implies an electro-optical transistor whose transconductance can be controlled both with the bias voltage (the horizontal axis) and the photon number. For instance, the bias voltage of 8 V and laser intensity of $40\,\mathrm{W\,cm^{-2}}$ leads to the transconductance of $10\,\mu\mathrm{S}$. Figure 5d shows the generated $I_f$ as the input ($V_s$) of $1\,V_{P-P}$ biased on $+8\,\mathrm{V}$ is applied, with and without the laser illumination ($40\,\mathrm{W\,cm^{-2}}$). Note that the flat port for Fig. 5b–d was short-circuited to solely study the coupled energy from the suspended port to the flat port.

The optical port in our studied device provides complete electrical isolation. Moreover, the electron-emitting surface is highly scalable and therefore is potentially capable of handling high power. These devices could also be used as photodetectors which can be tuned to a range of frequencies by adjusting the geometry of the surface. It can even be designed so that different frequencies resonate with different regions of the surface, providing a highly sensitive yet broadband response.

**Discussion**

Fig. 4a shows that the I–V curve of the flat port has asymmetry (versus the static bias polarity) which we suspect is associated with some physical asymmetry in the fabricated flat port. Figure 4b shows the responsivity of the flat port at $\lambda = 785\,\mathrm{nm}$ and with the bias voltage of $V_f = -1\,\mathrm{V}$ as the suspended port is open-circuited. Based on Fig. 4b, the current yield slope decreases significantly at laser intensities around $20\,\mathrm{W\,cm^{-2}}$ which indicates switching from the perturbative (multi-photon emission) to the strong-field light–matter interaction (tunnelling regime). In other words, the laser intensity of $20\,\mathrm{W\,cm^{-2}}$, in our design, creates pondermotive energy of electrons comparable with the biased gold work ionization energy (that is, Keldysh parameter is around unity)[18]. This typically requires laser intensities above $\mathrm{TW\,cm^{-2}}$ in the absence of any field enhancement caused by the laser–matter interaction. In Teichmann et al.[18], it is shown that a $1\,\mathrm{GW\,cm^{-2}}$ laser intensity suffices for photoemission if SPPs are excited. Here, we observe that LSPRs (which typically provide higher FE than traveling SPPs) along with a few $\mathrm{V\,\mu m^{-1}}$ static electric field decrease the laser intensity requirement for photoemission to the $\mathrm{W\,cm^{-2}}$ range. The wavelength dependence of the suspended port's response is shown in Fig. 4c. As we expected, electron photoemission has a peak at $\lambda = 785\,\mathrm{nm}$ which is consistent with the full-wave simulated result shown in Fig. 2b. It is worth mentioning that the emitted electrons through LSPR-enhanced photoemission are much more energetic than a conventional photoemission, which has been interpreted in terms of pondermotive acceleration[39–41]. It comes from the fact that localized electric fields of LSPRs are tightly bound to the metal surface, on the orders much smaller than the quiver amplitude of an electron. This leads to electrons traveling far from the metal surface before reversing direction during the second half-period of the laser temporal oscillation. Figure 4 also shows that the effect of the laser illumination and the static bias are comparable on the device's conductivity change. This is despite the fact that the laser's induced electric field on the metal surface (considering the FE) is smaller than the static bias. However, the static field does not penetrate into the metal and solely affects the potential barrier (besides increasing the electron density near the surface of

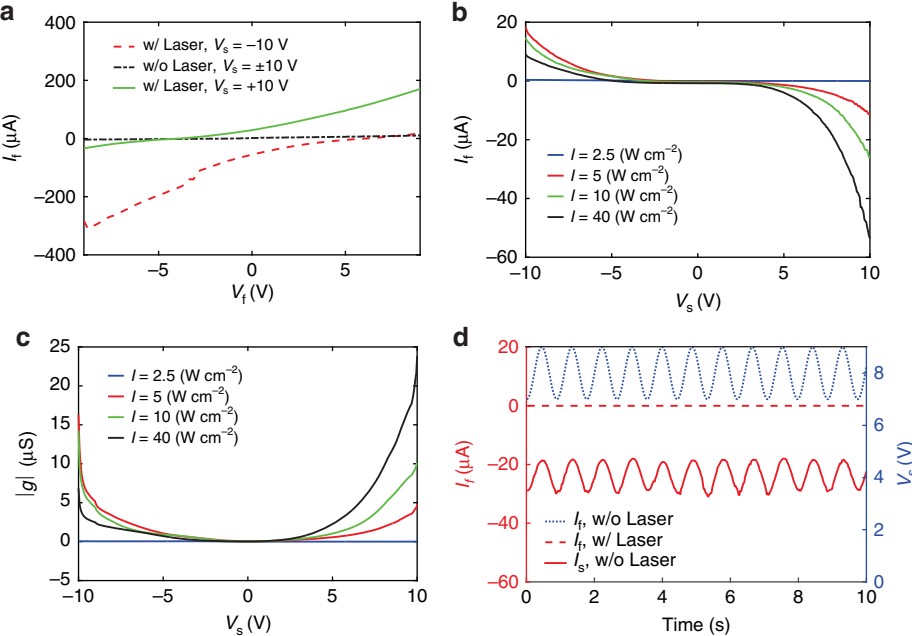

**Figure 5 | The mutual response between the two ports.** (**a**) I–V curves of the flat port as the suspended port is biased with different voltages, with (w/) and without (w/o) the laser illumination. laser intensity is $I = 5\,W\,cm^{-2}$, and subscripts f and s denote the flat and the suspended ports, respectively. (**b**) The induced current on the flat port as $V_f$ is fixed at zero and $V_s$ varies, (**c**) the small-signal transconductance ($|g|$) of the device ($V_f = 0$), (**d**) the induced sinusoidal current on the flat port due to the applied sinusoidal voltage on the suspended port ($V_f = 0$, $I = 40\,W\,cm^{-2}$).

the metal). On the contrary, photons can penetrate into the metal and increase the kinetic energy of electrons as well as affecting the potential barrier (conductivity of gold at 785 nm is around $37,000\,S\,m^{-1}$, leading to a skin depth of about 100 nm). In other words, an a.c. field at optical frequencies has larger effect than a static field with the same amplitude. LSPR excitation is a crucial matter in our design. As a comparison with a biased tip (without LSRP excitation), in Hommelhoff et al.[42], static and laser electric fields on the order of $0.5\,GV\,m^{-1}$ are applied to a Tungsten tip to emit electrons. By adding LSPR, Fig. 4 shows that electron emission can be achieved by applying a static electric field of $10\,MV\,m^{-1}$ along with a laser electric field less than $1\,MV\,m^{-1}$.

We may approximate the photoemission and field emission contributions to the current generation from Fig. 5b. It is evident that increasing the laser intensity is decreasing $I_f$ regardless of the $V_s$ polarity. This suggests that the photoemission current always has negative values in our measurements (this is consistent with the fact that electrons always leave the metal due to photoemission). On the other hand, the dection of the field emission current depends on the polarity of the applied static voltage. As a result, the net current in the flat port is the sum or difference of photoemission and electric field emission currents for positive or negative $V_s$, respectively. For instance, with the laser intensity of $I = 40\,W\,cm^{-2}$, applying $\pm 10\,V$ on the suspended port induces $-60$ or $10\,\mu A$ on the flat port, respectively. This leads to the conclusion that, for the specific laser intensity and bias voltage, the photoemission and field emission currents are $-25$ and $+35\,\mu A$, respectively. Simple calculations of the generated photoemission current and the photon energy at $\lambda = 785\,nm$ shows that photon to electron conversion rate is $\sim 5\%$. One should notice that dividing the current into the electric field emission and photoemission currents was a rough approximation. Applying the static field modifies the surface potential barrier, and applying the laser field changes both the potential barrier and the Fermi level of electrons in gold. Therefore, the obtained numbers based on the separation between the static bias

and the laser contributions may not have enough accuracy for design purposes.

In summary, we showed that photoemission enhanced by LSPRs and combined with electric field emission can liberate electrons from gold with the unprecedented laser intensities of $W\,cm^{-2}$. The fact that low bias voltages (under 10 V) and low power (a few mW) IR lasers can initiate and control the electron emission is very promising to design semiconductor-free devices with new opportunities to scale their capabilities (such as speed, power handling, and so on) to beyond what is limited by natural properties of semiconductors. The substrate in our design only supported the metallic structure and was not involved in the electric current flow. Events that would damage an ordinary semiconductor device (for example, over-voltage or radiation) would have little effect on a photoemission-based device.

## Methods

**Computer simulation.** A commercial finite element method code (ANSYS HFSS) was used to simulate the unit cell, and Johnson-Christy model was adopted for gold[43]. To simplify the simulations, mushrooms were set to have a smooth surface and rounded corners. This ensured that we avoided non-localities in the gold model and calculation singularities at the sharp corners[44]. As a result, the simulated field enhancement is a lower limit and, as will be shown later, the measured field enhancement will be larger.

**The experimental setup.** Supplementary Fig. 5 shows the measurement setup which includes a tunable Ti:Sapphire laser pumped with a 10 W green semiconductor laser. The output laser beam was passed through two beam samplers for wavelength and power measurements. An Ocean Optics spectrometer and a silicon photo-detector were used for wavelength and power measurements, respectively. The laser beam was passed through two beam samplers (for power and wavelength measurements) and a periscope before entering the vacuum chamber through a view port. The vacuum chamber was equipped with a vacuum pump, Ar gas inlet, pressure gauge, electrical feedthroughs and a customized imaging system observing the device from outside of the chamber. The fabricated devices were installed in standard dual in-line packages with a small piece of carbon conductive tape (typically used in SEM) and wire bonded using a ball bonder. Two source-meters (Keithley 2,400 and 2,410) with a common ground were used for full characterization of the three port devices. To measure the I–V

curves, the vacuum chamber was pumped down to $10^{-4}$ torr (our equipment limit). The negative electrodes of the two sourcemeters were connected to the optical table (ground), therefore two terminals of the device's ports were essentially connected. We also performed a few experiments to ensure that photoemission is the dominant process in our device (see Supplementary Figs 6 and 7 for more details).

**Fabrication.** A three-layer recipe was developed and optimized to perform the fabrication of clean mushrooms and air bridges. The first layer consisted of the gold ribbons on the substrate, the second layer included vias and the third layer comprised the mushroom caps. After cleaning the wafer with acetone, 180 nm $SiO_2$ was deposited on the wafer using plasma sputtering. Then, the first layer was patterned and fabricated using Ebeam lithography and Ebeam evaporation (70 nm Au on top of 10 nm Cr as the adhesion layer). Similarly, the second layer (vias) was fabricated using Ebeam lithography and Ebeam evaporation (250 nm Au). To fabricate the third layer (mushroom caps), photoresist (AZ1505) was spin-coated on the sample and was ashed with oxygen plasma down to the thickness of 200 nm, so that the tip of the vias were exposed. Then, a few nanometers of chromium was sputter coated on the photoresist to prevent it from mixing with the Ebeam resist. Next, Ebeam resist was coated on the sample (without any soft baking) and was patterned using Ebeam lithography. The samples were then ready after metallization (70 nm Au), lift off using acetone, chromium plasma etching and oxygen plasma cleaning.

**Raman spectroscopy/field enhancement measurement.** Experimental FEs were calculated from experimentally determined Raman FEs by comparing the enhanced spectra of thiophenol, a common SERS marker, to bulk Raman measurements of thiophenol and then normalizing to the respective number of Raman excited molecules. Because $FE = (E/E_0)^4$, where $(E/E_0)$ is the electric field enhancement, we can approximate the average field enhancement as $FE^{0.25}$. Thiophenol forms a self-assembled monolayer on gold surfaces. We performed an overnight thiophenol vapour phase deposition on the device. Excess thiophenol was removed by placing the device under vacuum for $>2$ h. Raman measurements were conducted on both bulk thiophenol and on the thiophenol monolayer coating the gold surface of the device. Both sets of measurements were carried out using the same measurement configuration. All data was collected using either a 785 nm diode laser, or a 633 nm HeNe laser, at powers of $<1$ mW, to ensure no desorption of the monolayer or morphological changes to the gold structure.

The FE was then determined using

$$FE = \left(\frac{I_{SERS}}{I_{Raman}}\right)\left(\frac{N_{Raman}}{N_{SERS}}\right), \tag{1}$$

in which $I$ is the measured bulk Raman or SERS Intensity, and $N$ is the number of molecules from which the Raman signal originates. $N_{Raman}$ was calculated using the density and molecular weight of bulk thiophenol along with laser focal volume. $N_{SERS}$ was calculated from the area percentage occupied by the gold structure, multiplied by the literature packing value for thiophenol self-assembled monolayers of 6.8 molecules per $nm^2$. The laser spot size was measured using the scanning knife-edge method[45]. Briefly, the laser was focused onto silicon under the same illumination conditions used for device measurements, and scanned over a cleaved edge in both the $X$ and $Y$ directions. The Raman intensity for the 520 cm$^{-1}$ vibrational mode of silicon was recorded at each point of the scan. The data were then fitted to error functions and the Gaussian beam waists derived. Laser focal depth was calculated by translating the Si along the $z$ axis and measuring the 520 cm$^{-1}$ Raman intensity. Data were fitted to a Gaussian and focal depth was determined as the integral ($-$ inf, inf) of the Gaussian. FE was calculated using the 999 cm$^{-1}$ vibrational mode of thiophenol because it displays low orientational dependence (that is, the Raman enhancement of this vibrational mode does not change substantially as a result of an average molecular ordering, as is the case with a self-assembled monolayer on a gold surface). In addition, it displays the highest bulk Raman signal and so gives us the most conservative FE calculation. Standard deviations were determined with measurements at $>15$ random points over the device surface.

**Data availability.** All important data generated or analysed during this study are included in this published article (and its Supplementary Information file). Further datasets are available from the corresponding author on reasonable request.

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

## Acknowledgements

The authors thank Shiva Piltan, and UC San Diego nanofabrication facility staff including Sean Parks, Larry Grissom, Ryan Anderson, Ivan Harris, and Xuekun Lu for the helpful discussions, and especially Maribel Montero for performing Ebeam lithography exposures. This work was funded by Defense Advanced Research Projects Agency (DARPA) through grant N00014-13-1-0618 and ONR DURIP through grant N00014-13-1-0655.

## Author contributions

D.S. proposed the idea and supervised the study. E.F. conceived and conducted the fabrication and experiments. T.J.D. performed and A.R.T. supervised the Ramon spectroscopy measurements. E.F. and D.S. analysed the results. E.F. wrote the manuscript. All authors reviewed the manuscript.

## Additional information

**Competing financial interests:** The authors declare no competing financial interests.

**Publisher's note**: 

