## [Peer review file · Nature Communications]

Reviewers' comments:

Reviewer #1 (Remarks to the Author):

This manuscript describes a novel type of device that is based on electron emission and that should be able to emulate microelectronic devices like transistors, switches or modulators with improved properties. The authors show that the device, with two electrical ports and one optical port, can control the current in one electrical port by both the voltage over the other port and a low incident laser intensity. The authors show that the current can be switched between a low and high level or be modulated.

The work in itself is very interesting and should warrant publication. However, in its current state I judge the manuscript not suitable for publication. The main reason is that the manuscript provides detailed experimental data but lacks in describing the physical mechanism behind the experimental observation. The authors do present sufficient data to support their claims.

In the introduction the authors describe the four main mechanisms that produce electron emission from a material (typically taken to be a metal). The first mechanism, thermionic emission, is properly described. The same is true for field emission. However, photoemission is not necessarily a multi-photon process and multiphoton absorption is not a defining property of photoemission. The authors should be more precise with the definition of the photoelectric effect and photoemission. The most general definition of the photoelectric effect used now-a-days is that absorption of a photon causes an electron to transit to a higher energy level (see e.g. Fundamentals of Photonics by Saleh and Teich). The photoelectric effect is divided into two forms, internal and external. The internal photoelectric effect typically leads to photoconductivity, while the external photoelectric effect leads to photoemission in which an electron leaves the material that absorbs one or more photons. Often, when authors refer to the photoelectric effect they implicitly refer to the external photoelectric effect, or photoemission. Photoemission is therefore synonymous with the external photoelectric effect. Furthermore, the authors suggest that, typically, extremely high light intensities are required for photoemission. However, that is not true. For example, photomultiplier tubes, that employ photocathodes to convert incident light into free electrons, are used for single photon counting applications. Also the remark that field emission requires high bias voltages of more than 100 V is not correct. So-called field emitter arrays (or Spindt arrays) require a gate voltage of only a few volts to turn on the emission. The text should be adjusted to provide the right description of these mechanisms.

My impression after reading the text is that authors assume that field enhancement (FE) (through LSPR) for the electric field of the incident light is the dominant contribution to the observed photoemission. I am not convinced by this. The FE is about a factor 30 as reported in the manuscript. A 5 W/cm² laser intensity leads to a field strength of about 60 V/cm. Therefore, one expects an electric field magnitude due to the laser light of about 2×10^5 V/m in the hotspots due to the FE. A lower estimate for the static electric field due to the bias voltage is about 10 V over a distance of about 120 nm, leading to a field strength of about 0.85×10^8 V/m. The distance is taken to be the shortest distance between adjacent rows of the suspended gold plates. From the unit cell (value for height h is missing in caption figure 2) I deduce that the distance between the suspended plate and bars of the flat port should not be different by a factor of 2. This means that the static electric field is larger by three orders of magnitude (two orders if the bias voltage is reduced to 1 V). Such a strong field will significantly modify the surface barrier potential and affect both field and photon-emission. (see e.g. Photoelectric field emission of electrons: Photon assisted tunneling by M. S. Sodha, A. Dixit, and S. Srivastava, Applied Physics Letters, 94, 251501 (2009); doi: 10.1063/1.3158595). This is true for the field between adjacent rows of suspended plates but also for the field between suspended plates and bars of the flat port if the voltage difference between these is similar. The bias voltage will determine from which (row of) plates we will have photoemission. Compare for example with a photocathode in an RF

accelerator. There, electrons will only be emitted and accelerated when the electric field associated with the RF field inside the accelerator is pointing in the right direction. For a field that is 180 degrees out of phase (reversal of bias voltage), no electron emission is observed).

The manuscript does not discuss the physics of the electron emission to any detail. (i) it does not provide any detail about the magnitude of the two emission processes (e.g. as given above), (ii) it does not describe how the bias voltages affect the emission process (e.g. does it change from which electrodes electrons are emitted) and (iii) do the authors assume that photoemission is only from the plates or does part of the light also reach (part of) the bar electrodes of the flat port? Further, in my opinion, the emission process becomes hybrid, in the sense that one cannot clearly distinguish between photon assisted field emission or field assisted photoemission. If the authors choose for one of the processes, they should make clear why the process should be named such.

Also, the work presented here resembles to some extent the work done on optical gated (laser induced or photo-assisted) field emission from so-called field emitter arrays (see e.g., doi: 10.1038/srep00915, 10.1116/1.1539060, 10.1063/1.2168031). This type of work is not discussed at all in the manuscript, while field emitter arrays have similar size (sub-micron length of unit cell) and similar microelectronic devices as suggested in this manuscript could be designed based on these nano-emitters.

Detailed comments:

- Always define symbols when they are first introduced. This is not the case in the manuscript.
- Units should not be in italic font as they appear to be at various locations.

p2 "such as the speed, power, wavelength, etc." Seems strange to mention wavelength here as this is typically associated with an optical wave, and here the authors are discussing existing electronic devices.

p2 "As and example" Type, should be "an"

p2 "atomic density of Ne" Subscript e suggests an electron density, while the text refers to atomic density. The two are only equal for 100 % single-ionized atoms.

p3 "(micro scale)" Remains unclear and may have different meaning for different readers. Please mention a characteristic scale length (nanometer?, micrometer?).

p3 "(normally in the ultraviolet range)" photocathodes operating with visible light exists (e.g. used in photo-multiplier tubes) and are sufficiently common for visible light to be considered common as well.

p3 "sperceded" Typo: "superseded"

p3 "semiconductor photo-resistors and photo-diodes" Not entirely accurate, photo-multiplier tubes are still around and provide one of the most sensitive detection of photons in certain spectral ranges.

p4 "yet they require bias voltages above 100 V" This is not accurate. Voltages required to turn on emission strongly dependent on the geometry used. There are designs for so-called field emitter arrays (or even a single emitter) that require only a few volts to turn on emission.

p4 "Typically, laser intensities on the order of TW/cm² (in the IR range) are required for photoemission" Not true, think for example of single photon counting done by photo-multiplier tubes (PMT). So this is definitely not typical. I would consider photoemission based on multi-photon absorption to be atypical instead of typical.

p4 "the same conclusions can be used at higher pressures for dimensions smaller than the mean free path of electrons (about μm in air)" Would this not require a reference?

p4 "unprecedented laser intensities of around W/cm²" W/cm² is just a unit, and does not quantify a level of intensity.

p4 "which can be seen as another advantage of the photoemission-based devices." I believe for most readers it is not clear what the advantage is. Make this explicit.

p4 "Figure 1 depicts our designed two-port device" It would be much clearer if the figure indicates to which parts the bias voltage will be applied (and which two electrodes share a common ground).

Further, should the optical field also not be considered as a port?

p4 "designed" Type: "designed"

p5, figure 2 "norm distribution" Insufficient description. Is this the magnitude of the electric field, or just the maximum of one of the components? Value for height h is missing. "field distribution" Of what and does this apply to all three figures? The specific wavelength should only apply to 2c, so make this clear to only include it in the description of 2c.

p5 "metallic inclusions, vertical gold posts topped with gold plates" Unclear if this is an explanation of what the inclusion is, or that it is part of the description of what makes up the unit cell. In both cases, the unit cell only contains one gold plate (disk) and one post on which the plate rests, so this should be singular.

p6 "included in Fig. 2," Unclear if this refers to the unit cell shown in Fig. 2 (Fig2a) or the result from the full wave simulation (Fig2c), or even Fig2b showing the resonance. Reference to individual figure parts should be made more explicit.

p6 "EF = 12" This is a relative measure. What is the absolute value? Under what conditions is this number obtained, i.e. what is the bias voltage and what is the incident laser intensity. Also the definition should be clearer. Does the incident electric field only refer to the laser field (as indicated by the name) or does this also include the bias field. Same counts for the maximum value, is this only due to enhancement of the laser field or does it also contain the bias field.

p6 "gap between mushrooms." It seems only natural to refer to Fig. 2c here.

p6 "the maximum static electric field (due to the bias), is superimposed with the laser-induced hot spot". The laser electric field is alternating. What about the relative magnitude and how does this effect electron emission?

p6 "half of the maximum FE," Again refer to Fig2c. From this figure, this is not immediately apparent. Also, this enhancement seems to come from the sharp edges of the geometry used in the simulation. A physical realization will have much smoother edges, so how much of this enhancement is expected to survive in an experimental realization?

p7 "average field enhancement of the samples" The use of average here is confusing. Just above, the authors mention that the maximum FE is 1.24 times the average FE reported in table 1. Do the authors refer here to the average maximum FE, i.e. maximum FE averaged over the 4 samples?

p7 "which provides substantial photoemission," Not sufficiently accurately formulated. The FE by itself is not sufficient to provide significant photoemission, e.g., Fig. 5b shows no significant photoemission at any incident laser intensity for zero bias, and no significant photoemission at any bias voltage for an incident intensity of 2.5 W/cm^2 . Also Fig 4a shows limited field emission that slightly increases with increasing voltage at the ports. So the FE alone is not sufficient for significant photoemission. The bias voltage is needed to suppress the surface barrier and this reduced surface barrier leads to significant photoemission at low laser intensities due to the FE. But this only leads to emission from alternate rows of plate where the surface barrier is reduced as for the other rows the surface barrier will be enhanced.

p8 table 1 "standard deviation (SD) and the average" I would reverse the order (in agreement with the column headings and also mention the two wavelengths in the caption. Typically, a table caption appears on top of the table. Check journal guidelines.

p8 "laser illumination". Mention wavelength and intensity here and not at the end of the discussion. Is the whole device illuminated? Has the laser spot been large enough to provide a homogeneous illumination? Does the laser beam have a Gaussian transverse fluence profile?

p8 "However, I_f has some negative value with laser illumination" Is this the asymmetry in I_f observed when positive and negative bias is applied? The authors should explain in more detail how the electron emission changes when the bias voltages change, e.g. from which electrodes are the electrons emitted?

p8 "The magnitude of this photoemission current increases from 100 nA to 800 nA as the laser intensity increases from 5 W/cm^2 to $S = 40 \text{ W/cm}^2$ " Information provided is incomplete. At which port is this photoemission current measured? And what are the voltages applied? Figure 4a shows much larger values for the current and current increase for laser off to laser on for nonzero applied voltage. Also, what fraction of the device is illuminated by the laser?

p8 "This can also explain the asymmetry in the I-V curve of the flat port in Fig. 4(a)" This is not immediately clear how a linearly changing photoemission current with incident laser intensity

explains the asymmetry. This should be better explained.

p9 "This ensures that photoemission is the prevalent mechanism in the device." As mentioned before, I am not sure how to label this emission process. I think we can't distinguish between field-assisted photo-emission or photon-assisted field emission. The authors should therefore reconsider this statement.

p9 "physical asymmetry in the fabricated flat port" Have the authors considered that by switching bias voltage from positive to negative, electrons may be emitted from different electrodes and therefore, (i) distance between emitting and receiving electrodes may vary and (ii) different surface roughness may affect the emission properties.

p9 "Figure 4 also includes the responsivity of the suspended port" Refer directly to the relevant subfigure. It is unclear what is meant with responsivity. Is that the emission current as a function of laser intensity (4b) or as a function of wavelength (4c). The next sentence would indicate the wavelength dependence and thus discussion of fig 4c. The following sentence jumps back to Fig. 4b. Authors need to make this clear and present the figures in the order that they are discussed in the text (or discuss the figures in the text in the order that they are presented).

p9 "in our design, creates ponderomotive energy of electrons comparable with the gold work function (i.e. Keldysh parameter is around unity)" I do not think that this is correctly stated. A Keldysh parameter equal to 1 corresponds to the ponderomotive energy of the electrons being equal the ionization energy of the material. However, due to the bias voltage applied, the surface barrier is significantly lowered and therefore the ionization energy is much lower. So, one should here not take the unperturbed work function of gold, but the work function perturbed by the bias voltage.

p9 "absence of any field enhancement caused by the laser-matter interaction." This is correct, but also the bias voltage plays an essential role. With zero bias voltage, the photoemission is much smaller than with bias voltage present (see fig. 4a). From the statement a little further in the text, it is clear that the bias voltage is essential to observe the transition from multi-photon-induced and tunneling emission.

p9 "This leads to electrons traveling far from the metal surface before reversing direction during the second half-period of the laser temporal oscillation." Is this argument valid in presence of the strong electric field due to the bias? This would only be true with zero bias voltage, and a non-zero bias voltage is essential for a large photoemission, as my simple estimate leads to the bias field to be the dominant field for the dynamics in the vacuum.

p9 "In all of the experiments of Fig. 4, one port is left open-circuited while measuring the other port. Also, except in Fig. 4(c), throughout this paper the wavelength is set to be $\lambda = 785$ nm." It would be much more useful for the reader to know this in advance before Fig. 4 is discussed in the text.

p10 figure 4. Why is this particular intensity chosen for fig. 4a? Why are these particular voltages used in presenting figs. 4b and 4c?

p10 "without any bias voltage on the suspended port" Does this mean an open circuit or zero volt as bias. These are not the same. The authors should clarify this.

p10 "For example, ..., respectively". Again a figure illustrating what voltage is applied to which electrodes would be useful in understanding the mechanism of the electron emission. For example, keeping the flat port at +10 V and switching the bias of the suspended port from -10 to +10 V will result in change in location from which the electrons are emitted.

p10 "This suggests us that the photoemission current always has negative values in our measurements (this is consistent with the fact that electrons always leave the metal due to photoemission)" It is true that emission is always due to electrons leaving the metal (both for photoemission and field emission). However, due to the bias voltage, a strong dc field is present which effects the surface barrier at the gold plates. In one row it enhances the barrier, while in the adjacent row the barrier is significantly reduced. Reversing the voltage on the suspended part, then results in switching of the row that produces photoemission. So the remark of the authors is not completely accurate. I am therefore not convinced by the subsequent discussion. As is clear from comments given above, I am not certain that the separation between photoemission and field emission is possible as the authors suggests it is.

p10 "Figure 5(d) shows" This is the only part in the manuscript where some dynamical response of

the device is given. However, the dynamical response is shown for a signal with a frequency of about 1 Hz. The authors claim that this device could have a much faster response than what traditional microelectronic devices can obtain. However, the dynamic response of the voltage and current through the electrical ports of the device are still limited by capacitance and inductance of the device. The authors should comment on this.

p11 "Fig. 8" This figure is not part of the manuscript. Do the authors refer to Fig 4a instead?

P11/12 The whole discussion on the contribution of the substrate to the changes in the observed conductance is somewhat out of place here (disrupts the flow of the text). I would suggest to move this part to the supplemental information and before discussing figure 4 mention in the text that the substrate (SiO₂ and Si) are not expected to contribute to changes in the conductance (see SI).

p13 "The fact that low bias voltages (under 10 volts) and low power (a few mW) IR lasers can initiate and control the electron emission". The authors should consider optical gating of field-emitter arrays as well. There, typically a gate voltage of only a few volts is need (below field emission threshold) and an incident laser pulse/beam initiates the emission. The authors could easily check what laser intensities are typically used.

p13 "Johnson-Christy model" Could the authors provide a reference?

p15 "laser spot size was calculated" The word calculated is a little confusing here. It would be simpler to write that the laser spot was measured using the scanning knife-edge method? (with a reference to that method). No value for the spot size is given, and it is not mentioned where in the beam line this measurement is made.

p15 "520 cm⁻¹ peak intensity" Of what? Why is this particular line chosen over the laser intensity itself? Same is true for the 999 cm⁻¹ peak intensity that is mentioned later in the text.

p15 "Field enhancement was calculated" This is confusing, as we are dealing here with a measured result. A calculation typically refers to a numeric model/theoretical calculation.

p15 "surface." what is the spot size used in each point?

Ref 9. Page/article number is missing

Ref 10/11 are not appropriate references to explain what what photoemission is.

Ref 12, 15. Page/article number is missing

Ref 21 seems incomplete

Supporting information:

- In figure S3, the authors use isolated and DC port, will in the text reference is made to flat and suspended port. Authors should use a single name for these ports.

- In figure S7, unit for voltage is missing in the legend. As it is not known for which of the configurations of figure S6 these measurements were done, I think that Figure S7 does not add anything to the statement already made in the text. This figure can be removed from the SI.

- Figure S8 in the text. It is mentioned that fabrication is very difficult. Does Fig. 8 represent a measurement of a successful manufacturing? The text should be more cleat in this respect. From the text I understood that the whole idea of using glass was to exclude a possible influence of the substrate on the conductance of the device. However, no conclusion in this respect is formulated here. Furthermore, if this is the figure 8 as mentioned in the manuscript, then the information supplied here does not agree with the information given in the text. Which is correct?

- Figure S8 itself: V_{DC} is not defined and in the main manuscript V_s is used. Be consistent with the symbols.

Reviewer #2 (Remarks to the Author):

In the manuscript, the authors study photoemission from engineered resonant surface to realize semiconductor-free microelectronic devices. They design structure with LSPR-based hot spots that significantly enhance plasmonic field and enable photoemission under low power diode laser. Overall, the study is interesting, but in my opinion, the probable impact of the paper is not high enough for Nature Communications.

In particular, photoemission in vacuum or gas is impractical: (1) switches and transistors require integration and deposition of other layers, which would not be possible in the proposed device; (2) photoemission at metal/semiconductor interface requires less energy than photoemission in vacuum, so the proposed device is energy-inefficient. Energy consumption is a key parameter for transistors and modulators; (3) realization of the device with vacuum or low-pressure gas environment is bulky and challenging from the design point of view.

Therefore, I do not recommend the publication.

Reviewer #3 (Remarks to the Author):

This is an excellent paper. Although the possible use of Surface Enhanced Raman Spectroscopy in nanostructured systems has been around for a while these authors have actually fabricated a nanoscale device that uses SERS to achieve electron tunneling emission into a partial vacuum where the mobility exceeds that of a semiconductor. And the enhancement achieved makes it possible to drive this system with a weak laser. The application is new and its integrability into nanoscale systems is important to the general scientific community. The paper is clear and well presented. Although this advance can be broken down into known issues - the paper in my opinion deserves publication and fanfare. A lot has been pulled together and the fabrication is new. There are a couple of quibbles that I wish to raise -not by way of questioning the importance or correctness of the paper but simply to pass on to the authors the response of a reviewer who has dealt with some of these issues. On page 4 of the MS 10V/um is cited as a small dc field. In my work with single crystals I have often run into surface flashover and worse at fields of 100,000V/cm. Have these authors discovered a new and better way of managing high dc fields? The authors claim a photon to electron conversion of 5%. My attempt at such a calculation gave closer to 100%. I had to make some assumptions which I thought were reasonable but which are probably wrong. Would the authors consider giving an example calculation of the 5% at a light intensity of 30W/cm².

Reviewers' comments:

Reviewer #1 (Remarks to the Author):

This manuscript describes a novel type of device that is based on electron emission and that should be able to emulate microelectronic devices like transistors, switches or modulators with improved properties. The authors show that the device, with two electrical ports and one optical port, can control the current in one electrical port by both the voltage over the other port and a low incident laser intensity. The authors show that the current can be switched between a low and high level or be modulated.

The work in itself is very interesting and should warrant publication. However, in its current state I judge the manuscript not suitable for publication. The main reason is that the manuscript provides detailed experimental data but lacks in describing the physical mechanism behind the experimental observation. The authors do present sufficient data to support their claims.

Thank you for your time. We tried to improve the physical description of the observations in the revised manuscript. As you also mentioned, the electron emission is due to the hybrid effect of photon interaction with electrons as well as the potential barrier change caused by the electric field (both static and laser driven). However, we had to make cautious statements about the exact physical mechanism based on our observation. We believe more studies, specifically designed to understand the physics, are needed for further explanation. Our goal in this work was to investigate the possibility of electron emission with low power IR lasers combined with the resonance effect and a static bias.

*In the introduction the authors describe the four main mechanisms that produce electron emission from a material (typically taken to be a metal). The first mechanism, thermionic emission, is properly described. The same is true for field emission. However, photoemission is not necessarily a multi-photon process and multiphoton absorption is not a defining property of photoemission. The authors should be more precise with the definition of the photoelectric effect and photoemission. The most general definition of the photoelectric effect used now-a-days is that absorption of a photon causes an electron to transit to a higher energy level (see e.g. *Fundamentals of Photonics* by Saleh and Teich). The photoelectric effect is divided into two forms, internal and external. The internal photoelectric effect typically leads to photoconductivity, while the external photoelectric effect leads to photoemission in which an electron leaves the material that absorbs one or more photons. Often, when authors refer to the photoelectric effect they implicitly refer to the external photoelectric effect, or photoemission. Photoemission is therefore synonymous with the external photoelectric effect.*

We agree that photoemission is not necessarily a multi-photon absorption process. We also agree that photoemission and photoelectric have the same nature (photon-matter interaction), and both the energy of a photon (its frequency) and the number of photons (the electric field intensity) are important for liberating an electron. Our classification was based on considering two limiting cases: 1) in the photon-driven electron emission, photon's energy is higher than the unperturbed work function of the atom, and the electron can be freed even with a single photon. This is the process which Hertz observed and is usually referred to as the photoelectric effect. 2) in the AC field-driven electron emission, the photon number is high (the electric field caused by the light is large) and the potential barrier perturbation causes several processes (e.g. electron tunneling, multiphoton absorption, over barrier transportation, etc.). Besides the two limiting cases, other

processes such as inter- and intra-band electron transitions inside the atom can occur as a result of light-matter interaction. We were referring to the AC field-driven electron emission as photoemission, for which high photon numbers are needed. Nonetheless, a different classification does not have any impact on the conclusions in this paper. We used the reviewer's suggested classification (internal/external photoelectric) in the revised manuscript.

The text is replaced with:

“The main mechanisms which extract electrons from a material (mostly metals) are: thermoionic emission, electric field emission, the photoelectric effect and photoemission. In a thermoionic emission process, electrons are transferred over the surface potential barrier of the metal (the work function) due to the added thermal energy. Most of the vacuum electronic devices, including vintage triodes and modern magnetrons, rely on thermoionic emission. However, thermionic emission requires cathode temperatures on the order of 1000K which makes it infeasible in micro-scale dimensions. Electric field emission, also called Fowler- Nordheim tunneling, is the process whereby electrons tunnel through the reduced work function due to the presence of a high electric field (typically static). This has been investigated for realizing cold cathode emitters to replace thermoionic emitters, since they do not require high power to thermally extract electrons from the surface, yet they require high bias voltages (e.g. close to 100V in [7]) [8, 9]. A general definition of the photoelectric effect is electron transition to a higher energy level due to excitation by a photon [10]. Depending on the photon's energy and number, electrons may be excited internally, leading to photoconductivity, or leave the metal, leading to electron emission. If the photon's energy is larger than the material's work function, photons can couple to electrons effectively and transfer them over the work function. This process, which was discovered by Heinrich Hertz in 1887 [11], is independent of the number of photons, and is used in designing vacuum/ gas-filled phototubes, and photo-multiplier tubes, some of which were eventually superseded by semiconductor photo-resistors and photo-diodes (photo-multiplier tubes are still in use). However, photons with an energy lower than the material's work function can also liberate electrons by either tunneling electrons through the potential barrier, or transferring them over the barrier (multi-photon absorption). This process requires a high number of photons, and is called photoemission in some works, as we do in the rest of this work [10, 12-16]. Unlike Hertz's experiments, the photon's energy in photoemission is less than the unperturbed metal work function, and the key factor is the laser-matter interaction (either strong-field or perturbative), caused by the nanolocalized electromagnetic field in the vicinity of metallic structures (such as sharp metallic nanotaperes). Typically, laser intensities on the order of $1\text{TW}=\text{cm}^2$ (in the IR range) are required for photoemission [17]. However, photoemission can be greatly enhanced by the excitation of collective electron modes of the metal, called surface plasmon polaritons (SPPs), and laser intensities on the order of $1\text{GW}=\text{cm}^2$ in the mid-IR have been shown to be enough for photoemission [18]. Nonetheless, the aforementioned electron emission mechanisms can be combined in order to increase the yield. For example, in the so-called thermo-field regime, the cathode temperature is elevated simultaneously with an applied electric field. In photomultiplier tubes [19] and field emitter arrays [7, 20, 21], a static bias is typically used to lower the potential barrier of the material, enabling it to emit electrons with light. This combination can provide great sensitivity (single photon in case of a photomultiplier tube) or high electron emission (bright electron beam in case of field emitter arrays). In this work, we add field emission to this approach by designing an engineered surface. Also, historically, these emission processes have been mostly studied for vacuum, and well-established theories exist to estimate their current densities. However, the same conclusions can be used at higher pressures for dimensions smaller than the mean free path of electrons (about $1\text{ }\mu\text{m}$ in air [22]).”

Furthermore, the authors suggest that, typically, extremely high light intensities are required for photoemission. However, that is not true. For example, photomultiplier tubes, that employ photocathodes to convert incident light into free electrons, are used for single photon counting applications.

Our statement was based on the assumption that photon's energy is smaller than the work function of the unbiased metal. We modified the text accounting the reviewer's point.

Also the remark that field emission requires high bias voltages of more than 100 V is not correct. So-called field emitter arrays (or Spindt arrays) require a gate voltage of only a few volts to turn on the emission. The text should be adjusted to provide the right description of these mechanisms.

We were also considering Spindt arrays as an example to make the statement. For example, In doi: 10.1038/srep00915 (which is also suggested by the reviewer), voltage of 95 V is applied to the array.

We adjusted the text to "...requires high bias voltages (e.g. close to 100V in \cite{mustonen2012efficient})."

My impression after reading the text is that authors assume that field enhancement (FE) (through LSPR) for the electric field of the incident light is the dominant contribution to the observed photoemission. I am not convinced by this. The FE is about a factor 30 as reported in the manuscript. A 5 W/cm² laser intensity leads to a field strength of about 60 V/cm. Therefore, one expects an electric field magnitude due to the laser light of about 2 10⁵ V/m in the hotspots due to the FE. A lower estimate for the static electric field due to the bias voltage is about 10 V over a distance of about 120 nm, leading to a field strength of about 0.85 10⁸ V/m. The distance is taken to be the shortest distance between adjacent rows of the suspended gold plates. From the unit cell (value for height h is missing in caption figure 2) I deduce that the distance between the suspended plate and bars of the flat port should not be different by a factor of 2. This means that the static electric field is larger by three orders of magnitude (two orders if the bias voltage is reduced to 1 V). Such a strong field will significantly modify the surface barrier potential and affect both field and photon-emission. (see e.g. Photoelectric field emission of electrons: Photon assisted tunneling by M. S. Sodha, A. Dixit, and S. Srivastava, Applied Physics Letters, 94, 251501 (2009); doi: 10.1063/1.3158595). This is true for the field between adjacent rows of suspended plates but also for the field between suspended plates and bars of the flat port if the voltage difference between these is similar. The bias voltage will determine from which (row of) plates we will have photoemission. Compare for example with a photocathode in an RF accelerator. There, electrons will only be emitted and accelerated when the electric field associated with the RF field inside the accelerator is pointing in the right direction. For a field that is 180 degrees out of phase (reversal of bias voltage), no electron emission is observed).

Yes, we also consider the combination of FE and the static bias responsible for the electron emission. We also had done the calculations the reviewer mentioned, and suspected that the static bias has the main contribution. However, our observations are not consistent with this assumption.

In fact, the reviewer's calculation lacks a point: the static field does not penetrate into the metal and solely affects the potential barrier (besides increasing the electron density near the surface of the metal). In the contrary, photons can penetrate into the metal and increase the kinetic energy of electrons as well as affecting the potential barrier (conductivity of gold at 785 nm is 37000 S/m, leading to a skin depth of about 100nm). In other words, AC field at optical frequencies has larger effect than a static field with the same amplitude. This is also consistent with our observation that laser effect (even with a smaller electric field) is more significant than the static bias. We added this statement to the revised manuscript.

The manuscript does not discuss the physics of the electron emission to any detail. (i) it does not provide any detail about the magnitude of the two emission processes (e.g. as given above), (ii) it does not describe how the bias voltages affect the emission process (e.g. does it change from which electrodes electrons are emitted) and (iii) do the authors assume that photoemission is only from the plates or does part of the light also reach (part of) the bar electrodes of the flat port? Further, in my opinion, the emission process becomes hybrid, in the sense that one cannot clearly distinguish between photon assisted field emission or field assisted photoemission. If the authors choose for one of the processes, they should make clear why the process should be named such.

We agree with the last statement of the reviewer, i.e. the emission process is hybrid, and we cannot clearly distinguish between photon-assisted and static-field assisted electron emission. This is clarified in the revised manuscript. Also, light interacts with both the flat and the suspended ports. Figure 4 shows how the conductivity of suspended and flat ports changes with the laser illumination. Also, from the simulation, Fig. 2, we can see that flat port also experiences FE almost half of the suspended port. Our conclusion is that both flat and suspended ports can contribute in the photoemission process.

Also, the work presented here resembles to some extent the work done on optical gated (laser induced or photo-assisted) field emission from so-called field emitter arrays (see e.g., doi: 10.1038/srep00915, 10.1116/1.1539060, 10.1063/1.2168031). This type of work is not discussed at all in the manuscript, while field

emitter arrays have similar size (sub-micron length of unit cell) and similar microelectronic devices as suggested in this manuscript could be designed based on these nano-emitters.

We added the discussion in the introduction. The references were also added.

Detailed comments:

-Always define symbols when they are first introduced. This is not the case in the manuscript.

-Units should not be in italic font as they appear to be at various locations.

Symbols have been fixed for (FE, Q, THz, and RF).

Units, subscripts, and superscripts have also been checked and fixed.

p2 "such as the speed, power, wavelength, etc." Seems strange to mention wavelength here as this is typically associated with an optical wave, and here the authors are discussing existing electronic devices.

Wavelength was removed.

p2 "As and example" Type, should be "an"

Fixed, thanks!

p2 "atomic density of Ne" Subscript e suggests an electron density, while the text refers to atomic density. The two are only equal for 100 % single-ionized atoms.

We meant atomic density. It is fixed in the revised version.

p3 "(micro scale)" Remains unclear and may have different meaning for different readers. Please mention a characteristic scale length (nanometer?, micrometer?).

Sub-micrometer scale.

p3 "(normally in the ultraviolet range)" photocathodes operating with visible light exists (e.g. used in photo-multiplier tubes) and are sufficiently common for visible light to be considered common as well.

You are right. The section was completely reworded.

p3 "sperceded" Typo: "superseded"

Thanks!

p3 "semiconductor photo-resistors and photo-diodes" Not entirely accurate, photo-multiplier tubes are still around and provide one of the most sensitive detection of photons in certain spectral ranges.

Photo-multiplier tubes were added to the statement as well. "...and photo-multipliers, some of which were eventually replaced by semiconductor photo-resistors and photo-diodes (photo-multiplier tubes are still in use).

p4 "yet they require bias voltages above 100 V" This is not accurate. Voltages required to turn on emission

strongly dependent on the geometry used. There are designs for so-called field emitter arrays (or even a single emitter) that require only a few volts to turn on emission.

Our reference for the statement was the Spindt array developed by SRI (doi: 10.1038/srep00915) in which 95 volts is required. Nonetheless, we modified the statement.

p4" Typically, laser intensities on the order of TW/cm² (in the IR range) are required for photoemission" Not true, think for example of single photon counting done by photo-multiplier tubes (PMT). So this is definitely not typical. I would consider photoemission based on multi-photon absorption to be atypical instead of typical.

Done. "A typical photoemission process based on multi-photon absorption requires laser intensities on the order of ..."

p4 "he same conclusions can be used at higher pressures for dimensions smaller than the mean free path of electrons (about μm in air)" Would this not require a reference?

A reference has been added
(<https://agenda.infn.it/getFile.py/access?contribId=27&resId=0&materialId=slides&confId=4542>).

p4 "unprecedented laser intensities of around W/cm²" W/cm² is just a unit, and does not quantify a level of intensity.

1 W/cm²

p4 "which can be seen as another advantage of the photoemission-based devices." I believe for most readers it is not clear what the advantage is. Make this explicit.

Done, "This provides an additional degree of freedom in designing photoemission-based devices, which can be seen as another advantage for them."

p4 "Figure 1 depicts our designed two-port device" It would be much clearer if the figure indicates to which parts the bias voltage will be applied (and which two electrodes share a common ground). Further, should the optical field also not be considered as a port?

Done. The ground electrode of each port is specified in the revised figure. Yes, the device also has an optical port. However, we still think it is more clear if Fig.1 only shows the electrical ports.

p4 "desinged" Type: "designed"

Thanks.

p5, figure 2 "norm distribution" Insufficient description. Is this the magnitude of the electric field, or just the maximum of one of the components? Value for height h is missing. "field distribution" Of what and does this apply to all three figures? The specific wavelength should only apply to 2c, so make this clear to only include it in the description of 2c.

By norm we meant Euclidean norm, which for the vector electric field is its magnitude, $|E|$. The figure caption is fixed, considering the rest of the reviewer's comments as well.

p5 "metallic inclusions, vertical gold posts topped with gold plates" Unclear if this is an explanation of what the inclusion is, or that it is part of the description of what makes up the unit cell. In both cases, the unit cell only contains one gold plate (disk) and one post on which the plate rests, so this should be singular.

Fixed.

p6 "included in Fig. 2," Unclear if this refers to the unit cell shown in Fig. 2 (Fig2a) or the result from the full wave simulation (Fig2c), or even Fig2b showing the resonance. Reference to individual figure parts should be made more explicit.

Done.

p6 "EF = 12" This is a relative measure. What is the absolute value? Under what conditions is this number obtained, i.e. what is the bias voltage and what is the incident laser intensity. Also the definition should be clearer. Does the incident electric field only refer to the laser field (as indicated by the name) or does this also include the bias field. Same counts for the maximum value, is this only due to enhancement of the laser field or does it also contain the bias field.

This value is the simulated ratio of the maximum to incident electric field. In the simulations, we used experimental values for the permittivity of gold, which do not include non-linear response of gold. In other words, the full-wave simulation results are independent of the incident power. The incident electric field only refers to the laser field (no bias is applied). The enhancement is also due to the laser (no bias is applied in Fig. 2). This has been clarified in the figure caption.

p6 "gap between mushrooms." It seems only natural to refer to Fig. 2c here.

Done.

p6 "the maximum static electric field (due to the bias), is superimposed with the laser-induced hot spot". The laser electric field is alternating. What about the relative magnitude and how does this effect electron emission?

We can claim the combination of the static and laser electric fields are causing the electron emission. For our device, we tried to repeat the measurements with different applied voltages and laser intensities (a summary of which is reported in Fig. 5). However, we are not quite confident to explain further the physics of the phenomena with the current device. We believe it requires a complete focused study to quantify and understand each field's (static and laser) contribution in the electron emission.

p6 "half of the maximum FE," Again refer to Fig2c. From this figure, this is not immediately apparent. Also, this enhancement seems to come from the sharp edges of the geometry used in the simulation. A physical realization will have much smoother edges, so how much of this enhancement is expected to survive in an experimental realization?

Done. You are right, the fabricated device did not have sharp edges of the flat port (in the simulation). However, we did not rely on the large enhancements seen at the edges of the ribbon in Fig. 3 (those points actually have FE of around the maximum). Instead, we were referring to the FE between the ribbons which have FE almost half of the maximum. Moreover, the fabricated device will also have surface roughness which further improves the FE (but is not included in the simulation).

p7 "average field enhancement of the samples" The use of average here is confusing. Just above, the authors mention that the maximum FE is 1.24 times the average FE reported in table 1. Do the authors refer here to the average maximum FE, i.e. maximum FE averaged over the 4 samples?

Yes, average maximum FE is correct, and is fixed.

p7 "which provides substantial photoemission," Not sufficiently accurate formulated. The FE by itself is not sufficient to provide significant photoemission, e.g., Fig. 5b shows no significant photoemission at any incident laser intensity for zero bias, and no significant photoemission at any bias voltage for an incident intensity of 2.5 W/cm². Also Fig 4a shows limited field emission that slightly increases with increasing voltage at the ports. So the FE alone is not sufficient for significant photoemission. The bias voltage is needed to suppress the surface barrier and this reduced surface barrier leads to significant photoemission at low laser intensities due the the FE. But this only leads to emission from alternate rows of plate where the surface barrier as reduced as for the other rows the surface barrier will be enhanced.

Yes, the corrected text reads "which provides substantial photoemission if combined with an applied static bias".

p8 table 1 "standard deviation (SD) and the average" I would reverse the order (in agreement with the column headings and also mention the two wavelengths in the caption. Typically, a table caption appears on top of the table. Check journal guidelines.

Done.

p8 "laser illumination". Mention wavelength and intensity here and not at the end of the discussion. Is the whole device illuminated? Has the laser spot been large enough to provide a homogeneous illumination? Does the laser beam have a Gaussian transverse fluence profile?

Done. We were using a Ti-Sapphire laser (3900S from Spectraphysics) with the beam diameter of 0.95mm which is much larger than the device area (20 by 20 microns). Therefore, uniform illumination of the device is a safe assumption. The laser output is a TM00 mode which implies a Gaussian beam.

p8 "However, If has some negative value with laser illumination" Is this the asymmetry in I_f observed when positive and negative bias is applied? The authors should explain in more detail how the electron emission changes when the bias voltages changes, e.g. from which electrodes are the electrons emitted?

p8 "The magnitude of this photoemission current increases from 100 nA to 800 nA as the laser intensity increases from 5 W/cm² to $S = 40$ W/cm²" Information provided is incomplete. At which port is this photoemission current measured? And what are the voltages applied? Figure 4a shows much larger values for the current and current increase for laser off to laser on for nonzero applied voltage. Also, what fraction of the device is illuminated by the laser?

p8 "This can also explain the asymmetry in the I-V curve of the flat port in Fig. 4(a)" This is not immediately clear how a linearly changing photoemission current with incident laser intensity explains the asymmetry. This should be better explained.

The text is reworded to clarify the point. Without applying a static voltage, the photoemission from the two inclusion of a port (either the flat or the suspended port) should be equal due to symmetry. Therefore, we expected a zero net current flow in the un-biased device's ports after laser illumination. However, I_{f} shows some current flow (negative) after laser illumination (even with $V_{\text{f}}=0$) which we suspect is due to some asymmetry in the flat port fabrication. The magnitude of this photoemission current (I_{f} while $V_{\text{f}}=0$) increases from 100 nA to 800 nA as the laser intensity increases from 5 W/cm^2 to $S=40\text{ W/cm}^2$. The possible asymmetry due to fabrication means one terminal of the flat port can emit electrons more efficiently than the other terminal. This can explain the asymmetry in the I-V curve of the flat port in Fig. 4(a) as well.

p9 "This ensures that photoemission is the prevalent mechanism in the device." As mentioned before, I am not sure how to label this emission process. I think we can't distinguish between field-assisted photo-emission or photon-assisted field emission. The authors should therefore reconsider this statement.

We agree, and we actually meant electron emission as opposed to gas ionization in air. The statement is fixed.

p9 "physical asymmetry in the fabricated flat port" Have the authors considered that by switching bias voltage from positive to negative, electrons may be emitted from different electrodes and therefore, (i) distance between emitting and receiving electrodes may vary and (ii) different surface roughness may affect the emission properties.

That is true. We also tried to extract as much information as possible by switching the applied voltages polarities, and we only reported the confident conclusions in the manuscript. In fact, one difficulty in understanding the physics of the device is the small distance between the inclusions. This allows strong electric field coupling between the four inclusions of a unit cell which makes it difficult to quantify each inclusion's contribution separately.

p9 "Figure 4 also includes the responsivity of the suspended port" Refer directly to the relevant subfigure. It is unclear what is meant with responsivity. Is that the emission current as a function of laser intensity (4b) or as a function of wavelength (4c). The next sentence would indicate the wavelength dependence and thus discussion of fig 4c. The following sentence jumps back to Fig. 4b. Authors need to make this clear and present the figures in the order that they are discussed in the text (or discuss the figures in the text in the order that they are presented).

Fixed.

p9 "in our design, creates pondermotive energy of electrons comparable with the gold work function (i.e. Keldysh parameter is around unity)" I do not think that this is correctly stated. A Keldysh parameter equal to 1 corresponds to the ponderomotive energy of the electrons being equal the ionization energy of the material. However, due to the bias voltage applied, the surface barrier is significantly lowered and therefore the ionization energy is much lower. So, one should here not take the unperturbed work function of gold, but the work function perturbed by the bias voltage.

You are right. This sentence was fixed by replacing "gold work function" with "biased gold ionization energy".

p9 "absence of any field enhancement caused by the laser-matter interaction." This is correct, but also the bias voltage plays an essential role. With zero bias voltage, the photoemission is much smaller than with bias voltage present (see fig. 4a). From the statement a little further in the text, it is clear that the bias voltage is essential to observe the transition from multi-photon-induced and tunneling emission.

p9 "This leads to electrons traveling far from the metal surface before reversing direction during the second half-period of the laser temporal oscillation." Is this argument valid in presence of the strong electric field due to the bias? This would only be true with zero bias voltage, and a non-zero bias voltage is essential for a large photoemission, as my simple estimate leads to the bias field to be the dominant field for the dynamics in the vacuum.

Yes, the argument was originally based on a zero bias voltage. However, with an applied static field, the argument is still valid for the electron-emitting terminal of the biased port. For this terminal, the static electric field helps pulling electrons out from inside the metal. In other words, with the applied static electric field, electrons travel even farther than without the static bias.

p9 "In all of the experiments of Fig. 4, one port is left open-circuited while measuring the other port. Also,

except in Fig. 4(c), throughout this paper the wavelength is set to be $\lambda = 785 \text{ nm}$." It would be much more useful for the reader to know this in advance before Fig. 4 is discussed in the text.

The statement is moved to before Fig. 4.

p10 figure 4. Why is this particular intensity chosen for fig. 4a? Why are these particular voltages used in presenting figs. 4b and 4c?

There was not a specific reason for choosing the laser intensities and bias voltages in Fig. 4 (they are simply random numbers). Although, we avoided laser intensities above 30 W/cm^2 to prevent heating the device.

p10 "without any bias voltage on the suspended port" Does this mean an open circuit or zero volt as bias. These are not the same. The authors should clarify this.

We meant open circuit, fixed in the text.

p10 "For example, ..., respectively". Again a figure illustrating what voltage is applied to which electrodes would be useful in understanding the mechanism of the electron emission. For example, keeping the flat port at +10 V and switching the bias of the suspended port from -10 to +10 V will result in change in location from which the electrons are emitted.

Done, Fig. 1 is updated.

p10 "This suggests us that the photoemission current always has negative values in our measurements (this is consistent with the fact that electrons always leave the metal due to photoemission)" It is true that emission is always due to electrons leaving the metal (both for photoemission and field emission). However, due to the bias voltage, a strong dc field is present which affects the surface barrier at the gold plates. In one row it enhances the barrier, while in the adjacent row the barrier is significantly reduced. Reversing the voltage on the suspended part, then results in switching of the row that produces photoemission. So the remark of the authors is not completely accurate. I am therefore not convinced by the subsequent discussion. As is clear from comments given above, I am not certain that the separation between photoemission and field emission is possible as the authors suggest it is.

We understand the reviewer's concern, and we are totally aware of the fact that separation between photoemission and electric field emission is not possible. We were only trying to provide a very rough approximation of the photoemission and electric field emission contributions, and subsequently provide a crude approximation of the photon to electron conversion. In the revised manuscript, we clarified this point further. However, if the reviewer/editor believes this part is misleading, we can modify this section and remove the approximations. This has been added:

"One should notice that dividing current into the electric field emission and photoemission currents was a rough approximation. Applying the static field modifies the surface potential barrier, and applying the laser field changes both the potential barrier and the Fermi level of electrons in gold. Therefore, the obtained numbers based on the separation between the static bias and the laser contributions may not have enough accuracy for design purposes."

p10 "Figure 5(d) shows" This is the only part in the manuscript where some dynamical response of the device is given. However, the dynamical response is shown for a signal with a frequency of about 1 Hz. The authors claim that this device could have a much faster response than what traditional microelectronic devices can obtain. However, the dynamic response of the voltage and current through the electrical ports of the device are still limited by capacitance and inductance of the device. The authors should comment on this.

Such low frequency was chosen due to the low reading rate of the sourcemeter (Keithely 2410) which is limited to a few hundreds of Hertz and we needed enough samples to generate smooth curves. However, we agree that the dynamic response of the designed device is limited by the inductance and capacitance (mostly) of the device. In fact, we had some evidences during measurement that the device speed is less than a MHz, but we were not sure about the test equipment and especially the device connectors and therefore we did not report it. Nonetheless, we did not intend to design an optimized high speed device in this work. Our intention was to show the possibility of realizing practical photoemission-based devices using combined low static voltage and low power lasers using resonant surfaces.

p11 "Fig. 8" This figure is not part of the manuscript. Do the authors refer to Fig 4a instead?

Yes, thanks

P11/12 The whole discussion on the contribution of the substrate to the changes in the observed conductance is somewhat out of place here (disrupts the flow of the text). I would suggest to move this part to the supplemental information and before discussing figure 4 mention in the text that the substrate (SiO₂ and Si) are not expected to contribute to changes in the conductance (see SI).

Done.

p13 "The fact that low bias voltages (under 10 volts) and low power (a few mW) IR lasers can initiate and control the electron emission". The authors should consider optical gating of field-emitter arrays as well. There, typically a gate voltage of only a few volts is need (below field emission threshold) and an incident laser pulse/beam initiates the emission. The authors could easily check what laser intensities are typically used.

Done.

p13 "Johnson-Christy model" Could the authors provide a reference?

Done.

p15 "laser spot size was calculated" The word calculated is a little confusing here. It would be simpler to write that the laser spot was measured using the scanning knife-edge method? (with a reference to that method). No value for the spot size is given, and it is not mentioned where in the beam line this measurement is made.

The laser spot size was measured by monitoring Raman intensity as the laser scans over a cleaved Si edge, the scanning knife-edge method (Ru, E.; Blackie, E.; Meyer, M.; Etchegoin, P. G. Surface Enhanced Raman Scattering Enhancement Factors: A Comprehensive Study. *J. Phys. Chem. C* **2007** 111, 13794). The laser spot size was measured at 1.6 μm x 117 μm (rectangular stripe diode) and is given on P7. The measurement is made at the focal plane of the Raman objective, with the same illumination conditions used for device characterization.

p15 "520 cm⁻¹ peak intensity" Of what? Why is this particular line chosen over the laser intensity itself? Same is true for the 999 cm⁻¹ peak intensity that is mentioned later in the text.

The 520 cm^{-1} peak is the Raman band resulting from the lattice phonon vibrational mode of Si. This peak is used because it is the strongest peak resulting from a knife-edge scan over cleaved Si. The laser intensity cannot be used in a Raman spectrometer as an edge filter is present to block the incident beam from the detector. The 999 cm^{-1} Raman peak for thiophenol is addressed in the next point.

p15 "Field enhancement was calculated" This is confusing, as we are dealing here with a measured result. A calculation typically refers to a numeric model/theoretical calculation.

The electric field enhancement (FE) cannot be measured directly, and thus must be calculated from the experimentally determined Raman enhancement factor (EF). Raman intensity is proportional to the intensity of incident light. When considering a SERS EF, the incident intensity is that of the plasmonically enhanced near-field. This near-field also enhances the Raman scattered light. If we assume the Raman shift is negligible compared to the plasmon resonance width, then this is reduced to $EF = (E/E_0)^4$, where (E/E_0) is the electric field enhancement. This is the equation with which we calculate the electric field enhancement from an experimentally determined EF. We use the 999 cm^{-1} Raman peak of thiophenol to determine the experimental EF because it has a low orientational dependence (i.e. the Raman enhancement of this vibrational mode does not change substantially as a result of an average molecular ordering, as is the case with a self-assembled monolayer on an Au surface). Specifics for the method of determining experimental EF are detailed in the methods section of the manuscript on P15.

p15 "surface." what is the spot size used in each point?

For the 785 nm laser excitation, the same $1.6\text{ }\mu\text{m} \times 16.95\text{ }\mu\text{m}$ laser spot is used as is used for spot size determination. For 633 nm excitation, we experimentally determined our spot size to be $1.2\text{ }\mu\text{m}$ in diameter.

P14 has been changed for clarification:

"Experimental FEs were calculated from experimentally determined Raman EFs by comparing the enhanced spectra of thiophenol, a common SERS marker, to bulk Raman measurements of thiophenol and then normalizing to the respective number of Raman excited molecules. Because $EF = (E/E_0)^4$, where (E/E_0) is the electric field enhancement, we can approximate the average field enhancement as $EF^{.25}$."

P15 has been revised to read as follows:

"The EF was then determined from the following equation

$$EF = \left(\frac{I_{SERS}}{I_{Raman}} \right) \left(\frac{N_{Raman}}{N_{SERS}} \right)$$

in which I is the measured bulk Raman or SERS Intensity, and N is the number of molecules from which the Raman signal originates. N_{Raman} was calculated using the density and molecular weight of bulk thiophenol along with laser focal volume. N_{SERS} was calculated from the area percentage occupied by the gold structure, multiplied by the literature packing value for thiophenol SAMs of $6.8\text{ molecules/nm}^2$. The laser spot size was measured using the scanning knife-edge method ^(Ref above). Briefly, the laser was focused onto silicon under the same illumination conditions used for device measurements, and scanned over a cleaved edge in both the X and Y directions. The Raman intensity for the 520 cm^{-1} vibrational mode of silicon was recorded at each point of the scan. The data was then fitted to error functions and the Gaussian beam waists derived. Laser focal depth was calculated by translating the Si along the z-axis and measuring the 520 cm^{-1} Raman intensity. Data was fitted to a Gaussian and focal depth was determined as the integral (-inf, inf) of the Gaussian. FE was calculated using the 999 cm^{-1} vibrational mode of thiophenol because it displays low orientational dependence (i.e. the Raman enhancement of this vibrational mode does not change substantially as a result of an average molecular ordering, as is the case with a self-assembled monolayer on an Au surface). In addition it displays the highest bulk Raman signal and so gives us the most conservative FE calculation. Standard deviations were determined with measurements at >15 random points over the device surface."

Ref 9. Page/article number is missing

Fixed.

Ref 10/11 are not appropriate references to explain what photoemission is.

Fixed. Refs. 10 and 11 in the revised manuscript are added.

Ref 12, 15. Page/article number is missing

Fixed.

Ref 21 seems incomplete

Fixed.

Supporting information:

- In figure S3, the authors use isolated and DC port, will in the text reference is made to flat and suspended port. Authors should use a single name for these ports.

Fixed.

- In figure S7, unit for voltage is missing in the legend. As it is not known for which of the configurations of figure S6 these measurements were done, I think that Figure S7 does not add anything to the statement already made in the text. This figure can be removed from the SI.

Units are added to the legend.

- Figure S8 in the text. It is mentioned that fabrication is very difficult. Does Fig. 8 represent a measurement of a successful manufacturing? The text should be more clear in this respect. From the text I understood that the whole idea of using glass was to exclude a possible influence of the substrate on the conductance of the device. However, no conclusion in this respect is formulated here. Furthermore, if this is the figure 8 as mentioned in the manuscript, then the information supplied here does not agree with the information given in the text. Which is correct?

Fig.8 is showing the results for a device fabricated on silicon (not glass). This has been clarified in the revised text.

- Figure S8 itself: V_{DC} is not defined and in the main manuscript V_s is used. Be consistent with the symbols.

Fixed.

Reviewer #2 (Remarks to the Author):

In the manuscript, the authors study photoemission from engineered resonant surface to realize semiconductor-free microelectronic devices. They design structure with LSPR-based hot spots that significantly enhance plasmonic field and enable photoemission under low power diode laser. Overall, the study is interesting, but in my opinion, the probable impact of the paper is not high enough for Nature Communications.

In particular, photoemission in vacuum or gas is impractical: (1) switches and transistors require integration and deposition of other layers, which would not be possible in the proposed device; (2) photoemission at metal/semiconductor interface requires less energy than photoemission in vacuum, so the proposed device is

energy-inefficient. Energy consumption is a key parameter for transistors and modulators; (3) realization of the device with vacuum or low-pressure gas environment is bulky and challenging from the design point of view.

We do not agree with the statement "...photoemission in vacuum or gas is impractical". There are numerous groups working on nano-scale vacuum electronic devices (which indeed work based on electron emission into vacuum and gas). In fact, there are several ongoing research programs on nano-scale vacuum electronic. For example, NASA has been working on nanoscale vacuum transistors since 2013 (http://www.nasa.gov/centers/ames/cct/office/cif/2013/nanoscale_vacuum.html), and DARPA has a program for developing integrated, microfabricated vacuum electronics (<http://www.darpa.mil/program/high-frequency-integrated-vacuum-electronics>)

Also, "switches and transistors require integration and deposition of other layers, which would not be possible in the proposed device;

The main proposed concept is the possibility of photoemission in micro-scale devices using the combination of a low power IR laser and a small (few volts) static bias. The device in this paper is designed to examine this possibility, and to some extent study its physics. We did not aim for optimizing the device for integration, or for a specific application. Although, the fact that the device is made of gold, fabricated on a silicon substrate (with SiO₂ isolation) and is fabricated using lithography steps, makes it suitable for integration with other semiconductor-based devices as well.

(2) photoemission at metal/semiconductor interface requires less energy than photoemission in vacuum, so the proposed device is energy-inefficient. Energy consumption is a key parameter for transistors and modulators

Again, the device was not optimized to compete with semiconductor transistors and modulators. Moreover, the key advantage of a vacuum device is its higher mobility of electrons (and hence higher speed) than its semiconductor counterparts.

(3) realization of the device with vacuum or low-pressure gas environment is bulky and challenging from the design point of view.

We agree that the design of such devices are challenging (not necessarily bulky though). However, they offer enough advantages which makes them worth considering (we refer the reviewer to the NASA and DARPA programs as examples). Moreover, as we mentioned on page 9, our experiments showed close results at air pressure too (which is explained by considering the mean free path of an electron in air). In other words, we did not claim that photo-emission based devices are necessarily vacuum electronic devices.

Therefore, I do not recommend the publication.

Reviewer #3 (Remarks to the Author):

This is an excellent paper. Although the possible use of Surface Enhanced Raman Spectroscopy in nanostructured systems has been around for a while these authors have actually fabricated a nanoscale device that uses SERS to achieve electron tunneling emission into a partial vacuum where the mobility exceeds that of a semiconductor. And the enhancement achieved makes it possible to drive this system with a weak laser. The application is new and its integrability into nanoscale systems is important to the general scientific community. The paper is clear and well presented. Although this advance can be broken down into known issues - the paper in my opinion deserves publication and fanfare. A lot has been pulled together and the fabrication is new.

There are a couple of quibbles that I wish to raise -not by way of questioning the importance or correctness of

the paper but simply to pass on to the authors the response of a reviewer who has dealt with some of these issues. On page 4 of the MS 10V/um is cited as a small dc field. In my work with single crystals I have often run into surface flashover and worse at fields of 100,000V/cm. Have these authors discovered a new and better way of managing high dc fields? The authors claim a photon to electron conversion of 5%. My attempt at such a calculation gave closer to 100%. I had to make some assumptions which I thought were reasonable but which are probably wrong. Would the authors consider giving an example calculation of the 5% at a light intensity of 30W/cm².

Thanks very much for your time and opinion about the paper. Our reference for citing 10 V/um as a small DC field was the Spindt cathode developed at SRI (see Anguero, V. M., and R. C. Adamo. "Space applications of spindt cathode field emission arrays." *6th Spacecraft Charging Technology*. 1998.) in which electric fields up to 200 V/um are applied to the cold cathode emitter. However, we agree that even 10 V/um is a large field intensity compared to many applications, and therefore we have not called it a small static field in the revised manuscript. We did not use any special methods for applying the static bias. Four large pads were placed around the device, and the pads were connected to the package's pins using wire bonding (ball bonder). The package's pins were then connected to a mini-smb connector which in turn was connected to a coaxial cable adaptor.

The photon to electron conversion rate is very dependent to the static bias on the inclusions. The 5% example was given based on Fig. 4(b) where the applied static voltage on the inclusions is small (1 volts). Based on Fig. 4(b), the light intensity of 30 W/cm² leads to a current flow of around 2.5 uA. The area of the device is 17um by 18um= 300e-8 cm². Therefore, the incident power on the device is 30W/cm²*300e-8 W=90 uW. Energy of each photon is $hf=6.62e-34*3e8/785e-9=2.5e-19$ Joules. Therefore, the incident photon rate on the device is $90e-6/2.5e-19=3.6e14$ per second. The generated current is about 2.5uA which is equivalent to the electron rate of $2.5e-6/1.6e-19=1.5e13$ per second. This leads to a photon conversion rate of about 4.1 percent.

This conversion rate increases significantly by increasing the static bias on the inclusions. For example, with the applied voltage of 10 volts, this rate increases almost 10 times (40%). However, it is an arguable approximation to distinguish between the field emitted and photoemitted currents at high applied static voltages. Therefore, we refrained to mention such high photon conversion rates in the manuscript, and only reported the rate at a low applied static voltage (i.e. 4.1%). It is worth mentioning that high absorption rates (up to 90%) are reported in previous works (e.g., *Opt. Lett.* **15**, 866 (1990); *Phys. Rev. B* **53**, 11193 (1996); *J. Phys.: Condens. Matter* **9**, 5765 (1997); *Phys. Rev. B* **79**, 085425 (2009)).

Reviewers' comments:

Reviewer #1 (Remarks to the Author):

This manuscript describes a novel type of device that is based on electron emission and that should be able to emulate microelectronic devices like transistors, switches or modulators with improved properties. The authors show that the device, with two electrical ports and one optical port, can control the current in one electrical port by both the voltage over the other port and a low incident laser intensity, using unprecedented low optical intensities. The authors show that the current can be switched between a low and high level or be modulated.

Although the route towards, e.g., a working transistor may be long with lots of technological challenges, the manuscript describes a novel and interesting concept that should be published.

The work is now well presented, based on sound experimental data using appropriate analysis methods. The conclusions are well supported by the experimental data. The manuscript is well written.

The authors now present a more balanced view of the physics behind the emission, where field enhancement, (static) bias voltage and the optical field all play a role. In my opinion, these improvements allow the manuscript to be published.

I therefore advise publication of the manuscript.

Reviewer #3 (Remarks to the Author):

2nd Report of Reviewer #3

This paper describes an experimental advance which combines 1) lowering of the barrier to photoemission with a geometrically focused external bias; and 2) use of a nanofabricated SERS structure which resonantly increases the electric field of imposed light; to achieve a switch that can be thrown with only $\sim 1\text{W}/\text{cm}^2$ of light. This key result is shown in Figure 4a. I propose that this type of system could be useful as a cathode source for accelerators and also as a source for HHG {higher harmonic generation}. So, I am not concerned with the [scientifically correct] qualms about energy efficiency stated by reviewer #2. Reviewer #1 is mostly concerned with shortcomings regarding the description of the physical mechanism at play. My perspective is that although advances 1,2) have been separately developed [for the geometric enhancement of PE see Hommelhoff PRL97,(2006) 247402; and for SERS see refs 25, 26, 27.] it is a first to combine both to achieve a 'device'. However, in reading the extensive analysis of reviewer #1 I did look back at a few experimental issues that I hadn't commented on previously. For instance Figure 2b shows a geometric enhancement of about a factor of 5 plus a resonant enhancement at 760nm of about a factor of 12.5 . Was it the goal of this research to achieve that extra factor of 2.5? And is it this factor which makes it possible to achieve switching at $\sim <5\text{W}/\text{cm}^2$ light. It would be useful if the authors had compared their SERS enhanced results to what could be obtained from biased tips- and explained why this factor of 2.5 is so key. Further--on page 11 the authors say "Without any bias voltage on the suspended port (open circuit), and with $V_f = 10\text{V}$ the laser illumination --". So from where to where is the V_f applied and why would one want an open circuit?

This reviewer supports publication of this paper because it broaches an interesting and widely applicable vision for new devices. However, the experiment should be clearly explained.

Reviewers' comments:

Reviewer #1 (Remarks to the Author):

This manuscript describes a novel type of device that is based on electron emission and that should be able to emulate microelectronic devices like transistors, switches or modulators with improved properties. The authors show that the device, with two electrical ports and one optical port, can control the current in one electrical port by both the voltage over the other port and a low incident laser intensity, using unprecedented low optical intensities. The authors show that the current can be switched between a low and high level or be modulated. Although the route towards, e.g., a working transistor may be long with lots of technological challenges, the manuscript describes a novel and interesting concept that should be published.

The work is now well presented, based on sound experimental data using appropriate analysis methods. The conclusions are well supported by the experimental data. The manuscript is well written.

The authors now present a more balanced view of the physics behind the emission, where field enhancement, (static) bias voltage and the optical field all play a role. In my opinion, these improvements allow the manuscript to be published.

I therefore advise publication of the manuscript.

Thank you for your time.

Reviewer #3 (Remarks to the Author):

2nd Report of Reviewer #3

This paper describes an experimental advance which combines 1) lowering of the barrier to photoemission with a geometrically focused external bias; and 2) use of a nanofabricated SERS structure which resonantly increases the electric field of imposed light; to achieve a switch that can be thrown with only $\sim 1\text{W}/\text{cm}^2$ of light. This key result is shown in Figure 4a. I propose that this type of system could be useful as a cathode source for accelerators and also as a source for HHG {higher harmonic generation}.

Thanks for mentioning this point. We added these applications, as well as photocathodes for free electron lasers, to the introduction section.

So, I am not concerned with the [scientifically correct] qualms about energy efficiency stated by reviewer #2. Reviewer #1 is mostly concerned with shortcomings regarding the description of the physical mechanism at play. My perspective is that although advances 1,2) have been separately developed [for the geometric enhancement of PE see Hommelhoff PRL97,(2006) 247402; and for SERS see refs 25, 26, 27.] it is a first to combine both to achieve a 'device'.

We added the reference Hommelhoff PRL97,(2006) 247402, as well.

However, in reading the extensive analysis of reviewer #1 I did look back at a few experimental issues that I hadn't commented on previously. For instance Figure 2b shows a geometric enhancement of about a factor of 5 plus a resonant enhancement at 760nm of about a factor of 12.5 . Was it the goal of this research to achieve that extra factor of 2.5? And is it this factor which makes it possible to achieve switching at $\sim <5\text{W}/\text{cm}^2$ light. It would be useful if the authors had compared their SERS enhanced results to what could be obtained from biased tips- and explained why this factor of 2.5 is so key.

Yes, one of the goals in this research was to maximize the electric field enhancement without adding too much fabrication complications (e.g. the feature size is kept >100 nm). The extra factor of 2.5 in the electric field enhancement is equivalent to a factor of 6.25 in the laser power, which is significant (e.g. compare a <200 mW laser with a 1 W laser) and helps to lower the laser power requirement. As a comparison with a biased tip (without LSRP), in Hommelhoff PRL97,(2006) 247402, bias electric fields and laser electric fields on the order of 0.5 GV/m are applied in order to emit electrons. By adding LSPR, we showed that electron emission can be achieved by applying a static electric field of 10 MV/m along with a laser electric field less than 1 MV/m.

Further--on page 11 the authors say "Without any bias voltage on the suspended port (open circuit), and with $V_f = 10\text{V}$ the laser illumination --". So from where to where is the V_f applied and why would one want an open circuit?

V_f was applied on the flat port, as clarified in Fig. 1. The suspended port was left open-circuited because, in Fig. 4(a), we intended to study each port's individual response (without any applied voltage on the other port). This was done merely to help us understand the physics of the device better, and there is no practical reason to leave a fabricated port open-circuited. This has been clarified in the revised manuscript.

This reviewer supports publication of this paper because it broaches an interesting and widely applicable vision for new devices. However, the experiment should be clearly explained.

Thanks again for your time!